# Learning Representations in Reinforcement Learning: An Information Bottleneck Approach

## Abstract

The information bottleneck principle in (Tishby et al., 2000) is an elegant and useful approach to representation learning. In this paper, we investigate the problem of representation learning in the context of reinforcement learning using the information bottleneck framework, aiming at improving the sample efficiency of the learning algorithms. We analytically derive the optimal conditional distribution of the representation, and provide a variational lower bound. Then, we maximize this lower bound with the Stein variational (SV) gradient method (originally developed in (Liu & Wang, 2016; Liu et al., 2017)). We incorporate this framework in the advantageous actor critic algorithm (A2C)(Mnih et al., 2016) and the proximal policy optimization algorithm (PPO) (Schulman et al., 2017). Our experimental results show that our framework can improve the sample efficiency of vanilla A2C and PPO significantly. Finally, we study the information bottleneck (IB) perspective in deep RL with the algorithm called mutual information neural estimation(MINE) (Belghazi et al., 2018). We experimentally verify that the information extraction-compression process also exists in deep RL and our framework is capable of accelerating this process. We also analyze the relationship between MINE and our method, through this relationship, we theoretically derive an algorithm to optimize our IB framework without constructing the lower bound.

## 1 Introduction

In training a reinforcement learning algorithm, an agent interacts with the environment, explores the (possibly unknown) state space, and learns a policy from the exploration sample data. In many cases, such samples are quite expensive to obtain (e.g., requires interactions with the physical environment). Hence, improving the sample efficiency of the learning algorithm is a key problem in RL and has been studied extensively in the literature. Popular techniques include experience reuse/replay, which leads to powerful off-policy algorithms (e.g., (Mnih et al., 2013; Silver et al., 2014; Van Hasselt et al., 2015; Nachum et al., 2018a; Espeholt et al., 2018)), and model-based algorithms (e.g., (Hafner et al., 2018; Kaiser et al., 2019)). Moreover, it is known that effective representations can greatly reduce the sample complexity in RL. This can be seen from the following motivating example: In the environment of a classical Atari game: Seaquest, it may take dozens of millions samples to converge to an optimal policy when the input states are raw images (more than 28,000 dimensions), while it takes less samples when the inputs are 128-dimension pre-defined RAM data(Sygnowski & Michalewski, 2016). Clearly, the RAM data contain much less redundant information irrelevant to the learning process than the raw images. Thus, we argue that an efficient representation is extremely crucial to the sample efficiency.

In this paper, we try to improve the sample efficiency in RL from the perspective of representation learning using the celebrated information bottleneck framework (Tishby et al., 2000). In standard deep learning, the experiments in (Shwartz-Ziv & Tishby, 2017) show that during the training process, the neural network first "remembers" the inputs by increasing the mutual information between the inputs and the representation variables, then compresses the inputs to efficient representation related to

the learning task by discarding redundant information from inputs (decreasing the mutual information between inputs and representation variables). We call this phenomena **"information extraction-compression process"**(information E-C process). Our experiments shows that, similar to the results shown in (Shwartz-Ziv & Tishby, 2017), we first (to the best of our knowledge) observe the information extraction-compression phenomena in the context of deep RL (we need to use MINE(Belghazi et al., 2018) for estimating the mutual information). This observation motivates us to adopt the information bottleneck (IB) framework in reinforcement learning, in order to accelerate the extraction-compression process. The IB framework is intended to explicitly enforce RL agents to learn an efficient representation, hence improving the sample efficiency, by discarding irrelevant information from raw input data. Our technical contributions can be summarized as follows:

1. We observe that the "information extraction-compression process" also exists in the context of deep RL (using MINE(Belghazi et al., 2018) to estimate the mutual information).

2. We derive the optimization problem of our information bottleneck framework in RL. In order to solve the optimization problem, we construct a lower bound and use the Stein variational gradient method developed in (Liu et al., 2017) to optimize the lower bound.

3. We show that our framework can accelerate the information extraction-compression process. Our experimental results also show that combining actor-critic algorithms (such as A2C, PPO) with our framework is more sample-efficient than their original versions.

4. We analyze the relationship between our framework and MINE, through this relationship, we theoretically derive an algorithm to optimize our IB framework without constructing the lower bound.

Finally, we note that our IB method is orthogonal to other methods for improving the sample efficiency, and it is an interesting future work to incorporate it in other off-policy and model-based algorithms.

## 2 Related Work

Information bottleneck framework was first introduced in (Tishby et al., 2000). They solve the framework by iterative Blahut Arimoto algorithm, which is infeasible to apply to deep neural networks. (Shwartz-Ziv & Tishby, 2017) tries to open the black box of deep learning from the perspective of information bottleneck, though the method they use to compute the mutual information is not precise. (Alemi et al., 2016) derives a variational information bottleneck framework, yet apart from adding prior target distribution of the representation distribution $P(Z|X)$, they also assume that $P(Z|X)$ itself must be a Gaussian distribution, which limits the capabilities of the representation function. (Peng et al., 2018) extends this framework to variational discriminator bottleneck to improve GANs(Goodfellow et al., 2014), imitation learning and inverse RL.

As for improving sample-efficiency, (Mnih et al., 2013; Van Hasselt et al., 2015; Nachum et al., 2018a) mainly utilize the experience-reuse. Besides experience-reuse, (Silver et al., 2014; Fujimoto et al., 2018) tries to learn a deterministic policy, (Espeholt et al., 2018) seeks to mitigate the delay of off-policy. (Hafner et al., 2018; Kaiser et al., 2019) learn the environment model. Some other powerful techniques can be found in (Botvinick et al., 2019).

State representation learning has been studied extensively, readers can find some classic works in the overview (Lesort et al., 2018). Apart from this overview, (Nachum et al., 2018b) shows a theoretical foundation of maintaining the optimality of representation space. (Bellemare et al., 2019) proposes a new perspective on representation learning in RL based on geometric properties of the space of value function. (Abel et al., 2019) learns representation via information bottleneck(IB) in imitation/apprenticeship learning. To the best of our knowledge, there is no work that intends to directly use IB in basic RL algorithms.

## 3 Preliminaries

A Markov decision process(MDP) is a tuple, $(\mathcal{X}, \mathcal{A}, \mathcal{R}, \mathcal{P}, \mu)$, where $\mathcal{X}$ is the set of states, $\mathcal{A}$ is the set of actions, $\mathcal{R} : \mathcal{X} \times \mathcal{A} \times \mathcal{X} \to \mathbb{R}$ is the reward function, $\mathcal{P} : \mathcal{X} \times \mathcal{A} \times \mathcal{X} \to [0, 1]$ is the transition probability function(where $P(X'|X, a)$ is the probability of transitioning to state $X'$ given that the previous state is $X$ and the agent took action $a$ in $X$), and $\mu : \mathcal{X} \to [0, 1]$ is the starting state distribution. A policy $\pi : \mathcal{X} \to \mathcal{P}(\mathcal{A})$ is a map from states to probability distributions over actions, with $\pi(a|X)$ denoting the probability of choosing action $a$ in state $X$.

In reinforcement learning, we aim to select a policy $\pi$ which maximizes $K(\pi) = \mathbb{E}_{\tau \sim \pi}[\sum_{t=0}^{\infty} \gamma^t \mathcal{R}(X_t, a_t, X_{t+1})]$, with a slight abuse of notation we denote $\mathcal{R}(X_t, a_t, X_{t+1}) = r_t$. Here $\gamma \in [0, 1)$ is a discount factor, $\tau$ denotes a trajectory $(X_0, a_0, X_1, a_1, ...)$. Define the state value function as $V^{\pi}(X) = \mathbb{E}_{\tau \sim \pi}[\sum_{t=0}^{\infty} \gamma^t r_t|X_0 = X]$, which is the expected return by policy $\pi$ in state $X$. And the state-action value function $Q^{\pi}(X, a) = \mathbb{E}_{\tau \sim \pi}[\sum_{t=0}^{\infty} \gamma^t r_t|X_0 = X, a_0 = a]$ is the expected return by policy $\pi$ after taking action $a$ in state $X$.

Actor-critic algorithms take the advantage of both policy gradient methods and value-function-based methods such as the well-known A2C(Mnih et al., 2016). Specifically, in the case that policy $\pi(a|X; \theta)$ is parameterized by $\theta$, A2C uses the following equation to approximate the real policy gradient $\nabla_{\theta} K(\pi) = \nabla_{\theta} \hat{J}(\theta)$:

$$\nabla_{\theta} \hat{J}(\theta) \approx \sum_{t=0}^{\infty} \nabla_{\theta}[\log \pi(a_t|X_t; \theta)(R_t - b(X_t)) + \alpha_2 H(\pi(\cdot|X_t))] = \sum_{t=0}^{\infty} \nabla_{\theta} \hat{J}(X_t; \theta) \quad (1)$$

where $R_t = \sum_{i=0}^{\infty} \gamma^i r_{t+i}$ is the accumulated return from time step $t$, $H(p)$ is the entropy of distribution $p$ and $b(X_t)$ is a baseline function, which is commonly replaced by $V^{\pi}(X_t)$.

A2C also includes the minimization of the mean square error between $R_t$ and value function $V^{\pi}(X_t)$. Thus in practice, the total objective function in A2C can be written as:

$$J(\theta) \approx \sum_{t=0}^{\infty} \log \pi(a_t|X_t; \theta)(R_t - V^{\pi}(X_t)) - \alpha_1 \|R_t - V^{\pi}(X_t)\|^2 + \alpha_2 H(\pi(\cdot|X_t)) = \sum_{t=0}^{\infty} J(X_t; \theta)$$
$$(2)$$

where $\alpha_1, \alpha_2$ are two coefficients.

In the context of representation learning in RL, $J(X_t; \theta)$(including $V^{\pi}(X_t)$ and $Q^{\pi}(X_t, a_t)$) can be replaced by $J(Z_t; \theta)$ where $Z_t$ is a learnable low-dimensional representation of state $X_t$. For example, given a representation function $Z \sim P_{\phi}(\cdot|X)$ with parameter $\phi$, define $V^{\pi}(Z_t; X_t, \phi)|_{Z_t \sim P_{\phi}(\cdot|X_t)} = V^{\pi}(X_t)$. For simplicity, we write $V^{\pi}(Z_t; X_t, \phi)|_{Z_t \sim P_{\phi}(\cdot|X_t)}$ as $V^{\pi}(Z_t)$.

## 4 Framework

### 4.1 Information Bottleneck in Reinforcement Learning

The information bottleneck framework is an information theoretical framework for extracting relevant information, or yielding a representation, that an input $X \in \mathcal{X}$ contains about an output $Y \in \mathcal{Y}$. An optimal representation of $X$ would capture the relevant factors and compress $X$ by diminishing the irrelevant parts which do not contribute to the prediction of $Y$. In a Markovian structure $X \to Z \to Y$ where $X$ is the input, $Z$ is representation of $X$ and $Y$ is the label of $X$, IB seeks an embedding distribution $P^{\star}(Z|X)$ such that:

$$P^{\star}(Z|X) = \arg \max_{P(Z|X)} I(Y, Z) - \beta I(X, Z) = \arg \max_{P(Z|X)} H(Y) - H(Y|Z) - \beta I(X, Z)$$
$$= \arg \max_{P(Z|X)} -H(Y|Z) - \beta I(X, Z) \quad (3)$$

for every $X \in \mathcal{X}$, which appears as the standard cross-entropy loss[1] in supervised learning with a MI-regularizer, $\beta$ is a coefficient that controls the magnitude of the regularizer.

Next we derive an information bottleneck framework in reinforcement learning. Just like the label $Y$ in the context of supervised learning as showed in (3), we assume the supervising signal $Y$ in RL to be the accurate value $R_t$ of a specific state $X_t$ for a fixed policy $\pi$, which can be approximated by an n-step bootstrapping function $Y_t = R_t = \sum_{i=0}^{n-2} \gamma^i r_{t+i} + \gamma^{n-1} V^\pi(Z_{t+n-1})$ in practice. Let $P(Y|Z)$ be the following distribution:

$$P(Y_t|Z_t) \propto \exp(-\alpha(R_t - V^\pi(Z_t))^2) \tag{4}$$

.This assumption is heuristic but reasonable: If we have an input $X_t$ and its relative label $Y_t = R_t$, we now have $X_t$'s representation $Z_t$, naturally we want to train our decision function $V^\pi(Z_t)$ to approximate the true label $Y_t$. If we set our target distribution to be $C \cdot \exp(-\alpha(R_t - V^\pi(Z_t))^2)$, the probability decreases as $V^\pi(Z_t)$ gets far from $Y_t$ while increases as $V^\pi(Z_t)$ gets close to $Y_t$.

For simplicity, we just write $P(R|Z)$ instead of $P(Y_t|Z_t)$ in the following context.

With this assumption, equation (3) can be written as:

$$P^\star(Z|X) = \arg \max_{P(Z|X)} \mathbb{E}_{X,R,Z \sim P(X,R,Z)}[\log P(R|Z)] - \beta I(X,Z)$$

$$= \arg \max_{P(Z|X)} \mathbb{E}_{X \sim P(X), Z \sim P(Z|X), R \sim P(R|Z)}[-\alpha(R - V^\pi(Z))^2] - \beta I(X,Z) \tag{5}$$

The first term looks familiar with classic mean squared error in supervisd learning. In a network with representation parameter $\phi$ and policy-value parameter $\theta$, policy loss $\hat{J}(Z;\theta)$ in equation(1) and IB loss in (5) can be jointly written as:

$$L(\theta,\phi) = \mathbb{E}_{X \sim P(X), Z \sim P_\phi(Z|X)}[\underbrace{\hat{J}(Z;\theta) + \mathbb{E}_R[-\alpha(R - V^\pi(Z;\theta))^2]]}_{J(Z;\theta)} - \beta I(X,Z;\phi) \tag{6}$$

where $I(X,Z;\phi)$ denotes the MI between $X$ and $Z \sim P_\phi(\cdot|X)$. Notice that $J(Z;\theta)$ itself is a standard loss function in RL as showed in (2). Finally we get the ultimate formalization of IB framework in reinforcement learning:

$$P_{\phi^*}(Z|X) = \arg \max_{P_\phi(Z|X)} \mathbb{E}_{X \sim P(X), Z \sim P_\phi(Z|X)}[J(Z;\theta)] - \beta I(X,Z;\phi) \tag{7}$$

The following theorem shows that if the mutual information $I(X,Z)$ of our framework and common RL framework are close, then our framework is near-optimality.

**Theorem** 1 (Near-optimality theorem). Policy $\pi^r = \pi_{\theta^r}$, parameter $\phi^r$, optimal policy $\pi^\star = \pi_{\theta^\star}$ and its relevant representation parameter $\phi^\star$ are defined as following:

$$\theta^r, \phi^r = \arg \min_{\theta,\phi} \mathbb{E}_{P_\phi(X,Z)}[\log \frac{P_\phi(Z|X)}{P_\phi(Z)} - \frac{1}{\beta}J(Z;\theta)] \tag{8}$$

$$\theta^\star, \phi^\star = \arg \min_{\theta,\phi} \mathbb{E}_{P_\phi(X,Z)}[-\frac{1}{\beta}J(Z;\theta)] \tag{9}$$

Define $J^{\pi^r}$ as $\mathbb{E}_{P_{\phi^r}(X,Z)}[J(Z;\theta^r)]$ and $J^{\pi^\star}$ as $\mathbb{E}_{P_{\phi^\star}(X,Z)}[J(Z;\theta^\star)]$. Assume that for any $\epsilon > 0$, $|I(X,Z;\phi^\star) - I(X,Z;\phi^r)| < \frac{\epsilon}{\beta}$, we have $|J^{\pi^r} - J^{\pi^\star}| < \epsilon$.

4.2 Target Distribution Derivation and Variational Lower Bound Construction

In this section we first derive the target distribution in (7) and then seek to optimize it by constructing a variational lower bound.

---

[1]Mutual information $I(X,Y)$ is defined as $\int dX dZ P(X,Z) \log \frac{P(X,Z)}{P(X)P(Z)}$, conditional entropy $H(Y|Z)$ is defined as $-\int dY dZ P(Y,Z) \log P(Y|Z)$. In a binary-classification problem, $-\log P(Y|Z) = -(1-Y)\log(1-\hat{Y}(Z)) - Y\log(\hat{Y}(Z))$.

We would like to solve the optimization problem in (7):

$$\max_{P_\phi(Z|X)} \mathbb{E}_{X\sim P(X),Z\sim P_\phi(Z|X)}[\underbrace{J(Z;\theta) - \beta\log P_\phi(Z|X)}_{L_1(\theta,\phi)} + \underbrace{\beta\log P_\phi(Z)}_{L_2(\theta,\phi)}] \tag{10}$$

Combining the derivative of $L_1$ and $L_2$ and setting their summation to 0, we can get that

$$P_\phi(Z|X) \propto P_\phi(Z)\exp(\frac{1}{\beta}J(Z;\theta)) \tag{11}$$

We provide a rigorous derivation of (11) in the appendix(A.2). We note that though our derivation is over the representation space instead of the whole network parameter space, the optimization problem (10) and the resulting distribution (11) are quite similar to the one studied in (Liu et al., 2017) in the context of Bayesian inference. However, we stress that our formulation follows from the information bottleneck framework, and is mathematically different from that in (Liu et al., 2017). In particular, the difference lies in the term $L_2$, which depends on the the distribution $P_\phi(Z \mid X)$ we want to optimize (while in (Liu et al., 2017), the corresponding term is a fixed prior).

The following theorem shows that the distribution in (11) is an optimal target distribution (with respect to the IB objective $L$). The proof can be found in the appendix(A.3).

**Theorem** 2. (Representation Improvement Theorem) Consider the objective function $L(\theta,\phi) = \mathbb{E}_{X\sim P(X),Z\sim P_\phi(Z|X)}[J(Z;\theta)] - \beta I(X,Z;\phi)$, given a fixed policy-value parameter $\theta$, representation distribution $P_\phi(Z|X)$ and state distribution $P(X)$. Define a new representation distribution: $P_{\hat\phi}(Z|X) \propto P_\phi(Z)\exp(\frac{1}{\beta}J(Z;\theta))$. We have $L(\theta,\hat\phi) \geq L(\theta,\phi)$.

Though we have derived the optimal target distribution, it is still difficult to compute $P_\phi(Z)$. In order to resolve this problem, we construct a variational lower bound with a distribution $U(Z)$ which is independent of $\phi$. Notice that $\int dZP_\phi(Z)\log P_\phi(Z) \geq \int dZP_\phi(Z)\log U(Z)$. Now, we can derive a lower bound of $L(\theta,\phi)$ in (6) as follows:

$$L(\theta,\phi) = \mathbb{E}_{X,Z}[J(Z;\theta) - \beta\log P_\phi(Z|X)] + \beta\int dZP_\phi(Z)\log P_\phi(Z)$$

$$\geq \mathbb{E}_{X,Z}[J(Z;\theta) - \beta\log P_\phi(Z|X)] + \beta\int dZP_\phi(Z)\log U(Z)$$

$$= \mathbb{E}_{X\sim P(X),Z\sim P_\phi(Z|X)}[J(Z;\theta) - \beta\log P_\phi(Z|X) + \beta\log U(Z)] = \hat{L}(\theta,\phi) \tag{12}$$

Naturally the target distribution of maximizing the lower bound is:

$$P_\phi(Z|X) \propto U(Z)\exp(\frac{1}{\beta}J(Z;\theta)) \tag{13}$$

### 4.3 Optimization by Stein Variational Gradient Descent

Next we utilize the method in (Liu & Wang, 2016)(Liu et al., 2017)(Haarnoja et al., 2017) to optimize the lower bound.

Stein variational gradient descent(SVGD) is a non-parametric variational inference algorithm that leverages efficient deterministic dynamics to transport a set of particles $\{Z_i\}_{i=1}^n$ to approximate given target distributions $Q(Z)$. We choose SVGD to optimize the lower bound because of its ability to handle unnormalized target distributions such as (13).

Briefly, SVGD iteratively updates the "particles" $\{Z_i\}_{i=1}^n$ via a direction function $\Phi^\star(\cdot)$ in the unit ball of a reproducing kernel Hilbert space (RKHS) $\mathcal{H}$:

$$Z_i \leftarrow Z_i + \epsilon\Phi^\star(Z_i) \tag{14}$$

where $\Phi^*(\cdot)$ is chosen as a direction to maximally decrease[2] the KL divergence between the particles' distribution $P(Z)$ and the target distribution $Q(Z) = \frac{\hat{Q}(Z)}{C}$($\hat{Q}$ is unnormalized

---

[2]In fact, $\Phi^*$ is chosen to maximize the directional derivative of $F(P) = -\mathbb{D}_{KL}(P||Q)$, which appears to be the "gradient" of $F$

distribution, $C$ is normalized coefficient) in the sense that

$$\Phi^\star \leftarrow \arg\max_{\phi \in \mathcal{H}}\{-\frac{d}{d\epsilon}\mathbb{D}_{KL}(P_{[\epsilon\phi]}||Q) \quad s.t. \quad \|\Phi\|_{\mathcal{H}} \leq 1\} \tag{15}$$

where $P_{[\epsilon\Phi]}$ is the distribution of $Z + \epsilon\Phi(Z)$ and $P$ is the distribution of $Z$. (Liu & Wang, 2016) showed a closed form of this direction:

$$\Phi(Z_i) = \mathbb{E}_{Z_j \sim P}[\mathcal{K}(Z_j, Z_i)\nabla_{\hat{Z}}\log\hat{Q}(\hat{Z})\mid_{\hat{Z}=Z_j} + \nabla_{\hat{Z}}\mathcal{K}(\hat{Z}, Z_i)\mid_{\hat{Z}=Z_j}] \tag{16}$$

where $\mathcal{K}$ is a kernel function(typically an RBF kernel function). Notice that $C$ has been omitted.

In our case, we seek to minimize $\mathbb{D}_{KL}(P_\phi(\cdot|X)||\frac{U(\cdot)\exp(\frac{1}{\beta}J(\cdot;\theta))}{C})\mid_{C=\int dZU(Z)\exp(\frac{1}{\beta}J(Z;\theta))}$, which is equivalent to maximize $\hat{L}(\theta, \phi)$, the greedy direction yields:

$$\Phi(Z_i) = \mathbb{E}_{Z_j \sim P_\phi(\cdot|X)}[\mathcal{K}(Z_j, Z_i)\nabla_{\hat{Z}}(\frac{1}{\beta}J(\hat{Z};\theta) + \log U(\hat{Z}))\mid_{\hat{Z}=Z_j} + \nabla_{\hat{Z}}\mathcal{K}(\hat{Z}, Z_i)\mid_{\hat{Z}=Z_j}] \tag{17}$$

In practice we replace $\log U(\hat{Z})$ with $\zeta \log U(\hat{Z})$ where $\zeta$ is a coefficient that controls the magnitude of $\nabla_{\hat{Z}}\log U(\hat{Z})$. Notice that $\Phi(Z_i)$ is the greedy direction that $Z_i$ moves towards $\hat{L}(\theta, \phi)$'s target distribution as showed in (13)(distribution that maximizes $\hat{L}(\theta, \phi)$). This means $\Phi(Z_i)$ is the gradient of $\hat{L}(Z_i, \theta, \phi)$: $\frac{\partial\hat{L}(Z_i,\theta,\phi)}{\partial Z_i} \propto \Phi(Z_i)$.

Since our ultimate purpose is to update $\phi$, by the chain rule, $\frac{\partial\hat{L}(Z_i,\theta,\phi)}{\partial\phi} \propto \Phi(Z_i)\frac{\partial Z_i}{\partial\phi}$. Then for $\hat{L}(\theta, \phi) = \mathbb{E}_{P_\phi(X,Z)}[\hat{L}(Z, \theta, \phi)]$:

$$\frac{\partial\hat{L}(\theta, \phi)}{\partial\phi} \propto \mathbb{E}_{X \sim P(X), Z_i \sim P_\phi(\cdot|X)}[\Phi(Z_i)\frac{\partial Z_i}{\partial\phi}] \tag{18}$$

$\Phi(Z_i)$ is given in equation(17). In practice we update the policy-value parameter $\theta$ by common policy gradient algorithm since:

$$\frac{\partial\hat{L}(\theta, \phi)}{\partial\theta} = \mathbb{E}_{P_\phi(X,Z)}[\frac{\partial J(Z;\theta)}{\partial\theta}] \tag{19}$$

and update representation parameter $\phi$ by (18).

### 4.4 Verify the information E-C process with MINE

This section we verify that the information E-C process exists in deep RL with MINE and our framework accelerates this process.

Mutual information neural estimation(MINE) is an algorithm that can compute mutual information(MI) between two high dimensional random variables more accurately and efficiently. Specifically, for random variables X and Z, assume $T$ to be a function of $X$ and $Z$, the calculation of $I(X, Z)$ can be transformed to the following optimization problem:

$$I(X, Z) = \max_T \mathbb{E}_{P(X,Z)}[T] - \log(\mathbb{E}_{P(X)\otimes P(Z)}[\exp^T]) \tag{20}$$

The optimal function $T^\star(X, Z)$ can be approximated by updating a neural network $T(X, Z; \eta)$.

With the aid of this powerful tool, we would like to visualize the mutual information between input state $X$ and its relative representation $Z$: Every a few update steps, we sample a batch of inputs and their relevant representations $\{X_i, Z_i\}_{i=1}^n$ and compute their MI with MINE, every time we train MINE(update $\eta$) we just shuffle $\{Z_i\}_{i=1}^n$ and roughly assume the shuffled representations $\{Z_i^{\text{shuffled}}\}_{i=1}^n$ to be independent with $\{X_i\}_{i=1}^n$:

$$I(X, Z) \approx \max_\eta \frac{1}{n}\sum_{i=1}^n[T(X_i, Z_i; \eta)] - \log(\frac{1}{n}\sum_{i=1}^n[\exp^{T(X_i, Z_i^{\text{shuffled}};\eta)}]) \tag{21}$$

Figure(1) is the tensorboard graph of mutual information estimation between $X$ and $Z$ in Atari game Pong, x-axis is update steps and y-axis is MI estimation. More details and results can be found in appendix(A.6) and (A.7). As we can see, in both A2C with our framework and common A2C, the MI first increases to encode more information from inputs("remember" the inputs), then decreases to drop irrelevant information from inputs("forget" the useless information). And clearly, our framework extracts faster and compresses faster than common A2C as showed in figure(1)(b).

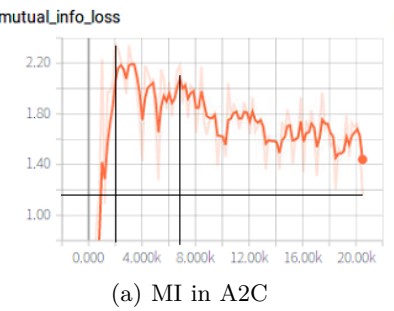

(a) MI in A2C

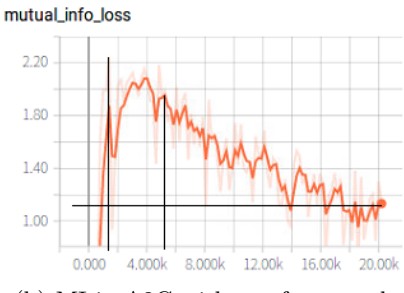

(b) MI in A2C with our framework

Figure 1: Mutual information visualization in Pong

After completing the visualization of MI with MINE, we analyze the relationship between our framework and MINE. According to (Belghazi et al., 2018), the optimal function $T^*$ in (20) goes as follows:

$$\exp^{T^*(X,Z;\eta)} = C\frac{P_\phi(X,Z)}{P(X)P_\phi(Z)} \quad s.t. \quad C = \mathbb{E}_{P(X)\otimes P_\phi(Z)}[\exp^{T^*}] \tag{22}$$

Combining the result with Theorem(2), we get:

$$\exp^{T^*(X,Z;\eta)} = C\frac{P_\phi(Z|X)}{P_\phi(Z)} \propto \exp(\frac{1}{\beta}J(Z;\theta)) \tag{23}$$

Through this relationship, we theoretically derive an algorithm that can directly optimize our framework without constructing the lower bound, we put this derivation in the appendix(A.5).

## 5 Experiments

In the experiments we show that our framework can improve the sample efficiency of basic RL algorithms(typically A2C and PPO). Our anonymous code can be found in https://github.com/AnonymousSubmittedCode/SVIB. Other results can be found in last two appendices.

In A2C with our framework, we sample $Z$ by a network $\phi(X,\epsilon)$ where $\epsilon \sim \mathcal{N}(\cdot;0,0.1)$ and the number of samples from each state $X$ is 32, readers are encouraged to take more samples if the computation resources are sufficient. We set the IB coefficient as $\beta = 0.001$. We choose two prior distributions $U(Z)$ of our framework, the first one is uniform distribution, apparently when $U(Z)$ is the uniform distribution, $\nabla_{\hat{Z}}\log U(\hat{Z})|_{\hat{Z}=Z}$ can be omitted. The second one is a Gaussian distribution, which is defined as follows: for a given state $X_i$, sample a batch of $\{Z_j^i\}_{j=1}^{n=32}$, then: $U(Z) = \mathcal{N}(Z;\mu = \frac{1}{n}\sum_{j=1}^n Z_j^i, \sigma^2 = \frac{1}{n}\sum_{j=1}^{n}(Z_j^i - \mu)^2)$.

We also set $\zeta$ as $0.005\|\nabla_{\hat{Z}}\frac{1}{\beta}J(\hat{Z};\theta)/\nabla_{\hat{Z}}\log U(\hat{Z})\||_{\hat{Z}=Z}$ to control the magnitude of $\nabla_{\hat{Z}}\log U(\hat{Z})|_{\hat{Z}=Z}$. Following (Liu et al., 2017), the kernel function in (17) we used is the Gaussian RBF kernel $\mathcal{K}(Z_i, Z_j) = \exp(-\|Z_i - Z_j\|^2/h)$ where $h = med^2/2\log(n+1)$, $med$ denotes the median of pairwise distances between the particles $\{Z_j^i\}_{i=1}^n$. As for the hyper-parameters in RL, we simply choose the default parameters in A2C of Openai-baselines(https://github.com/openai/baselines/tree/master/baselines/a2c). In summary, we implement the following four algorithms:

**A2C with uniform SVIB**: Use $\phi(X, \epsilon)$ as the embedding function, optimize by our framework(algorithm(A.4)) with $U(Z)$ being uniform distribution.

**A2C with Gaussian SVIB**: Use $\phi(X, \epsilon)$ as the embedding function, optimize by our framework(algorithm(A.4)) with $U(Z)$ being Gaussian distribution.

**A2C**:Regular A2C in Openai-baselines with $\phi(X)$ as the embedding function.

**A2C with noise**(For fairness):A2C with the same embedding function $\phi(X, \epsilon)$ as A2C with our framework.

Figure(2)(a)-(e) show the performance of four A2C-based algorithms in 5 gym Atari games. We can see that A2C with our framework is more sample-efficient than both A2C and A2C with noise in nearly all 5 games.

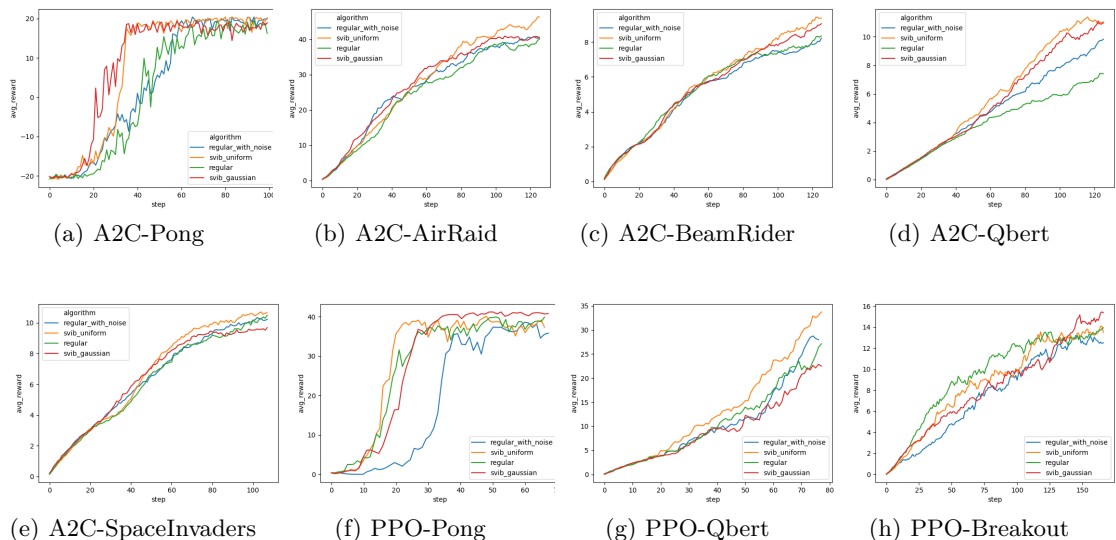

(a) A2C-Pong     (b) A2C-AirRaid     (c) A2C-BeamRider     (d) A2C-Qbert

(e) A2C-SpaceInvaders     (f) PPO-Pong     (g) PPO-Qbert     (h) PPO-Breakout

Figure 2: (a)-(e) show the performance of four A2C-based algorithms, x-axis is time steps(2000 update steps for each time step) and y-axis is the average reward over 10 episodes, (f)-(h) show the performance of four PPO-based algorithms, x-axis is time steps(300 update steps for each time step). We make exponential moving average of each game to smooth the curve(In PPO-Pong, we add 21 to all four curves in order to make exponential moving average). We can see that our framework improves sample efficiency of basic A2C and PPO.

Notice that in SpaceInvaders, A2C with Gaussian SVIB is worse. We suspect that this is because the agent excessively drops information from inputs that it misses some information related to the learning process. There is a more detailed experimental discussion about this phenomena in appendix(A.7) . We also implement four PPO-based algorithms whose experimental settings are same as A2C except that we set the number of samples as 26 for the sake of computation efficiency. Results can be found in the in figure(2)(f)-(h).

## 6   Conclusion

We study the information bottleneck principle in RL: We propose an optimization problem for learning the representation in RL based on the information-bottleneck framework and derive the optimal form of the target distribution. We construct a lower bound and utilize Stein Variational gradient method to optimize it. Finally, we verify that the information extraction and compression process also exists in deep RL, and our framework can accelerate this process. We also theoretically derive an algorithm based on MINE that can directly optimize our framework and we plan to study it experimentally in the future work.

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

## A  Appendix

### A.1  Proof of Theorem 1

**Theorem**. (Theorem 1 restated)Policy $\pi^r = \pi_{\theta^r}$, parameter $\phi^r$, optimal policy $\pi^\star = \pi_{\theta^\star}$ and its relevant representation parameter $\phi^\star$ are defined as following:

$$\theta^r, \phi^r = \arg\min_{\theta,\phi} \mathbb{E}_{P_\phi(X,Z)}[\log \frac{P_\phi(Z|X)}{P_\phi(Z)} - \frac{1}{\beta}J(Z;\theta)] \tag{24}$$

$$\theta^\star, \phi^\star = \arg\min_{\theta,\phi} \mathbb{E}_{P_\phi(X,Z)}[-\frac{1}{\beta}J(Z;\theta)] \tag{25}$$

Define $J^{\pi^r}$ as $\mathbb{E}_{P_{\phi^r}(X,Z)}[J(Z;\theta^r)]$ and $J^{\pi^\star}$ as $\mathbb{E}_{P_{\phi^\star}(X,Z)}[J(Z;\theta^\star)]$. Assume that for any $\epsilon > 0$, $|I(X,Z;\phi^\star) - I(X,Z;\phi^r)| < \frac{\epsilon}{\beta}$, we have $|J^{\pi^r} - J^{\pi^\star}| < \epsilon$. Specifically, in value-based algorithm, this theorem also holds between expectation of two value functions.

**Proof**. From equation(24) we can get:

$$I(X,Z;\phi^\star) - \frac{1}{\beta}J^{\pi^\star} \geq I(X,Z;\phi^r) - \frac{1}{\beta}J^{\pi^r} \tag{26}$$

From equation(25) we can get:

$$-\frac{1}{\beta}J^{\pi^r} \geq -\frac{1}{\beta}J^{\pi^\star} \tag{27}$$

These two equations give us the following inequality:

$$\beta(I(X,Z;\phi^\star) - I(X,Z;\phi^r)) \geq J^{\pi^\star} - J^{\pi^r} \geq 0 \tag{28}$$

According to the assumption, naturally we have:

$$|J^{\pi^r} - J^{\pi^\star}| < \epsilon \tag{29}$$

Notice that if we use our IB framework in value-based algorithm, then the objective function $J^\pi$ can be defined as:

$$J^\pi = V^\pi = (1-\gamma)^{-1} \int_X dX d^\pi(X) R^\pi(X)$$

$$= (1-\gamma)^{-1} \int_X dX d^\pi(X) [\int_Z dZ P_\phi(Z|X) R^\pi(Z)] \tag{30}$$

where $R^\pi(Z) = \int_{X \in \{X' : \phi(X')=Z\}} dX R^\pi(X)$ and $d^\pi$ is the discounted future state distribution, readers can find detailed definition of $d^\pi$ in the appendix of (Chen et al., 2018). We can get:

$$|V^{\pi^r} - V^{\pi^\star}| < \epsilon \tag{31}$$

## A.2 Target Distribution Derivation

We show the rigorous derivation of the target distribution in (11).

Denote $P$ as the distribution of $X$, $P_\phi^Z(Z) = P_\phi(Z)$ as the distribution of $Z$. We use $P_\phi$ as the short hand notation for the conditional distribution $P_\phi(Z|X)$. Moreover, we write $L(\theta, \phi) = L(\theta, P_\phi)$ and $\langle p, q \rangle_X = \int dX p(X) q(X)$. Notice that $P_\phi^Z(Z) = \langle P(\cdot), P_\phi(Z|\cdot) \rangle_X$. Take the functional derivative with respect to $P_\phi$ of the first term $L_1$:

$$\left\langle \frac{\delta L_1(\theta, P_\phi)}{\delta P_\phi}, \Phi \right\rangle_{XZ} = \int dZ dX \frac{\delta L_1(\theta, P_\phi(Z|X))}{\delta P_\phi(Z|X)} \Phi(Z,X) = \left[ \frac{d}{d\epsilon} L_1(\theta, P_\phi + \epsilon\Phi) \right]_{\epsilon=0}$$

$$= \left[ \frac{d}{d\epsilon} \int dX P(X) \left\langle P_\phi(\cdot|X) + \epsilon\Phi(\cdot,X), J(\cdot;\theta) - \beta \log(P_\phi(\cdot|X) + \epsilon\Phi(\cdot,X)) \right\rangle_Z \right]_{\epsilon=0}$$

$$= \int dX P(X) \left[ \left\langle \Phi(\cdot,X), J(\cdot;\theta) - \beta \log P_\phi(\cdot|X) \right\rangle + \left\langle P_\phi(\cdot|X), -\beta \frac{\Phi(\cdot,X)}{P_\phi(\cdot|X)} \right\rangle_Z \right]$$

$$= \left\langle P(\cdot)[J(\cdot;\theta) - \beta \log P_\phi(\cdot|\cdot) - \beta], \Phi(\cdot,\cdot) \right\rangle_{XZ}$$

Hence, we can see that

$$\frac{\delta L_1(\theta, P_\phi)}{\delta P_\phi(Z|X)} = P(X)[J(Z;\theta) - \beta \log P_\phi(Z|X) - \beta].$$

Then we consider the second term. By the chain rule of functional derivative, we have that

$$\frac{\delta L_2(\theta, P_\phi)}{\delta P_\phi(Z|X)} = \left\langle \frac{\delta L_2(\theta, P_\phi)}{\delta P_\phi^Z(\cdot)}, \frac{\delta P_\phi^Z(\cdot)}{\delta P_\phi(Z|X)} \right\rangle_{\hat{Z}} = \beta \left\langle 1 + \log P_\phi^Z(\cdot), \frac{\delta P_\phi^Z(\cdot)}{\delta P_\phi(Z|X)} \right\rangle_{\hat{Z}}$$

$$= \beta \int d\hat{Z}(1 + \log P_\phi^Z(\hat{Z})) \delta(\hat{Z} - Z) P(X) = \beta P(X)(1 + \log P_\phi^Z(Z)) \tag{32}$$

Combining the derivative of $L_1$ and $L_2$ and setting their summation to 0, we can get that

$$P_\phi(Z|X) \propto P_\phi(Z) \exp(\frac{1}{\beta} J(Z;\theta)) \tag{33}$$

## A.3 Proof of Theorem 2

**Theorem**. (Theorem 2 restated) For $L(\theta, \phi) = \mathbb{E}_{X \sim P(X), Z \sim P_\phi(Z|X)}[J(Z;\theta)] - \beta I(X, Z; \phi)$, given a fixed policy-value parameter $\theta$, representation distribution $P_\phi(Z|X)$ and state distribution $P(X)$, define a new representation distribution: $P_{\hat{\phi}}(Z|X) \propto P_\phi(Z) \exp(\frac{1}{\beta} J(Z;\theta))$, we have $L(\theta, \hat{\phi}) \geq L(\theta, \phi)$.

**Proof**. Define $I(X)$ as:

$$I(X) = \int_Z dZ P_{\hat{\phi}}(Z|X) = \int_Z dZ P_\phi(Z) \exp(\frac{1}{\beta} J(Z; \theta)) \tag{34}$$

$$L(\theta, \hat{\phi}) = \mathbb{E}_X \{ \mathbb{E}_{Z \sim P_{\hat{\phi}}(Z|X)}[J(Z; \theta)] - \beta \mathbb{E}_{Z \sim P_{\hat{\phi}}(Z|X)}[\log \frac{P_\phi(Z) \exp(\frac{1}{\beta} J(Z; \theta))}{I(X) P_{\hat{\phi}}(Z)}] \}$$

$$= \mathbb{E}_X \{ \beta \mathbb{E}_{Z \sim P_{\hat{\phi}}(Z|X)}[\log I(X)] - \beta \mathbb{E}_{Z \sim P_{\hat{\phi}}(Z|X)}[\log \frac{P_\phi(Z)}{P_{\hat{\phi}}(Z)}] \}$$

$$= \beta \mathbb{E}_X[\log I(X)] - \beta \mathbb{E}_{X, Z \sim P_{\hat{\phi}}(X, Z)}[\log \frac{P_\phi(Z)}{P_{\hat{\phi}}(Z)}]$$

$$= \beta \mathbb{E}_X[\log I(X)] - \beta \mathbb{E}_{Z \sim P_{\hat{\phi}}(Z)}[\log \frac{P_\phi(Z)}{P_{\hat{\phi}}(Z)}]$$

$$= \beta \mathbb{E}_{X \sim P(X)}[\log I(X)] + \beta \mathbb{D}_{KL}(P_{\hat{\phi}}(Z) || P_\phi(Z)) \tag{35}$$

$$L(\theta, \phi) = \mathbb{E}_X \{ \beta \mathbb{E}_{Z \sim P_\phi(Z|X)}[\log \exp(\frac{1}{\beta} J(Z; \theta))] + \beta \mathbb{E}_{Z \sim P_\phi(Z|X)} \log \frac{P_\phi(Z)}{P_\phi(Z|X)} \}$$

$$= \mathbb{E}_X \{ \beta \mathbb{E}_{Z \sim P_\phi(Z|X)}[\log \frac{P_\phi(Z) \exp(\frac{1}{\beta} J(Z; \theta))}{P_\phi(Z|X) I(X)}] + \beta \log I(X) \}$$

$$= \beta \mathbb{E}_X[\log I(X)] + \beta \mathbb{E}_{X \sim P(X), Z \sim P_\phi(Z|X)}[\log \frac{P_{\hat{\phi}}(Z|X)}{P_\phi(Z|X)}]$$

$$= \beta \mathbb{E}_{X \sim P(X)}[\log I(X)] - \beta \mathbb{E}_{X \sim P(X)}[\mathbb{D}_{KL}(P_\phi(Z|X) || P_{\hat{\phi}}(Z|X))] \tag{36}$$

$$L(\theta, \hat{\phi}) - L(\theta, \phi) = \beta \mathbb{D}_{KL}(P_{\hat{\phi}}(Z) || P_\phi(Z)) + \beta \mathbb{E}_{X \sim P(X)}[\mathbb{D}_{KL}(P_\phi(Z|X) || P_{\hat{\phi}}(Z|X))] \tag{37}$$

According to the positivity of the KL-divergence, we have $L(\theta, \hat{\phi}) \geq L(\theta, \phi)$.

## A.4 Algorithm

---
**Algorithm 1** Information-bottleneck-based state abstraction in RL
---
$\theta, \phi \leftarrow$ initialize network parameters
$\beta, \zeta \leftarrow$ initialize hyper-parameters in (17)
$\epsilon \leftarrow$ learning rate
$M \leftarrow$ number of samples from $P_\phi(\cdot|X)$
**repeat**
    Draw a batch of data $\{X_t, a_t, R_t, X_{t+1}\}_{t=1}^n$ from environment
    **for** each $X_t \in \{X_t\}_{t=1}^n$ **do**
        Draw M samples $\{Z_i^t\}_{i=1}^M$ from $P_\phi(\cdot|X_t)$
    **end for**
    Get the batch of data $\mathcal{D} = \{X_t, \{Z_i^t\}_{i=1}^M, a_t, R_t, X_{t+1}\}_{t=1}^n$
    Compute the representation gradients $\nabla_\phi L(\theta, \phi)$ in $\mathcal{D}$ according to (18)
    Compute the RL gradients $\nabla_\theta L(\theta, \phi)$ in $\mathcal{D}$ according to (19)
    Update $\phi$: $\phi \leftarrow \phi + \epsilon \nabla_\phi L(\theta, \phi)$
    Update $\theta$: $\theta \leftarrow \theta + \epsilon \nabla_\theta L(\theta, \phi)$
**until** Convergence
---

## A.5 Integrate MINE to our framework

MINE can also be applied to the problem of minimizing the MI between Z and X where Z is generated by a neural network $P_\phi(\cdot|X)$:

$$I^\star(X, Z) = \min_\phi \max_\eta \mathbb{E}_{P_\phi(X, Z)}[T(X, Z; \eta)] - \log(\mathbb{E}_{P(X) \otimes P_\phi(Z)}[\exp^{T(X, Z; \eta)}]) \tag{38}$$

Apparently $I^\star(X, Z)$ is 0 without any constraints, yet if we use MINE to optimize our IB framework in (6), $I^\star(X, Z)$ might not be 0. With the help of MINE, like what people did in supervised learning, the objective function in (6) can be written as:

$$L(\theta, \phi, \eta) = \min_\eta \{ \underbrace{\mathbb{E}_{P_\phi(X,Z)}[J(Z;\theta)]}_{\text{RL loss term}} - \beta \mathbb{E}_{P_\phi(X,Z)}[T(X, Z; \eta)]$$

$$+ \beta \log(\mathbb{E}_{P(X) \otimes P_\phi(Z)}[\exp^{T(X,Z;\eta)}]) \} \tag{39}$$

The key steps to optimize $L(\theta, \phi, \eta)$ is to update $\theta, \phi, \eta$ iteratively as follows:

$$\eta_{t+1} \leftarrow \arg\min_{\eta_t} \underbrace{-\mathbb{E}_{P_{\phi_t}(X,Z)}[T(X, Z; \eta_t)] + \log(\mathbb{E}_{P(X) \otimes P_{\phi_t}(Z)}[\exp^{T(X,Z;\eta_t)}])}_{\text{mutual information term}} \tag{40}$$

$$\theta_{t+1}, \phi_{t+1} \leftarrow \arg\max_{\theta_t, \phi_t} L(\theta_t, \phi_t, \eta_{t+1}) \tag{41}$$

Yet in our experiment, these updates of $\theta, \phi, \eta$ did not work at all. We suspect it might be caused by the unstability of $T(X, Z; \eta)$. We have also found that the algorithm always tends to push parameter $\phi$ to optimize the mutual information term to 0 regardless of the essential RL loss $J(Z; \theta)$: In the early stage of training process, policy $\pi$ is so bad that the agent is unable to get high reward, which means that the RL loss is extremely hard to optimize. Yet the mutual information term is relatively easier to optimize. Thus the consequence is that these updates tend to push parameter $\phi$ to optimize the mutual information term to 0 in the early training stage.

However, MINE's connection with our framework makes it possible to optimize $T(X, Z; \eta)$ in a more deterministic way and put the RL loss into mutual information term.

According to equation(23), $T(X, Z; \eta) = \frac{1}{\beta} J(Z; \theta) + C'$, this implies that we could introduce another function $\hat{T}$ in place of $T(X, Z; \eta)$ for the sake of variance reduction:

$$T(X, Z; \eta) \leftarrow \frac{1}{\beta} J(Z; \theta) + \hat{T}(X, Z; \eta) \tag{42}$$

in the sense that parameter $\eta$ only needs to approximate the constant $C'(T)$, the optimization steps turn out to be:

$$\theta_{t+1}, \phi_{t+1}, \eta_{t+1} \leftarrow \arg\max_{\theta_t, \phi_t} \arg\min_{\eta_t} \{ -\beta \mathbb{E}_{P_{\phi_t}(X,Z)}[\hat{T}(X, Z; \eta_t)] +$$

$$\beta \log(\mathbb{E}_{P(X) \otimes P_{\phi_t}(Z)}[\exp^{\hat{T}(X,Z;\eta_t) + \frac{1}{\beta} J(Z;\theta_t)}]) \} \tag{43}$$

This algorithm has the following three potential advantages in summary:

1. We're able to optimize $T(X, Z; \eta)$ in a more deterministic way instead of the form like equation(40), which is hard to converge in reinforcement learning.

2. We prevent excessive optimization in mutual information term by putting RL loss $J(Z; \theta)$ into this term.

3. We're able to directly optimize the IB framework without constructing a variational lower bound.

Here we just analyze and derive this algorithm theoretically, implementation and experiments will be left as the future work.

Notice that we say if we directly optimize our IB framework with MINE, one of the problems is that the function $T$ might be unstable in RL, yet in section(4.4) and experiments(A.6), we directly use MINE to visualize the MI. This is because when we optimize our framework, we need to start to update $T$ every training step. While when we visualize the MI, we start to update $T$ every 2000 training steps. Considering the computation efficiency, every time we start to update $T$ when we use MINE to optimize our framework, we must update $T$ in one step or a few steps. While when visualizing the MI, we update $T$ in 256 steps. Besides, we reset parameter $\eta$ every time we begin to update $T$ when we visualize the MI, clearly we can't do this when optimizing our framework since it's a min-max optimization problem.

A.6  Study the information-bottleneck perspective in RL

Now we introduce the experimental settings of MI visualization. And we show that the agent in RL usually tends to follow the information E-C process.

We compare the MI($I(X, Z)$) between A2C and A2C with our framework. Every 2000 update steps(2560 frames each step), we re-initialize the parameter $\eta$, then sample a batch of inputs and their relevant representations $\{X_i, Z_i = \phi(X_i, \epsilon)\}_{i=1}^n, n = 64$, and compute the MI with MINE. The learning rate of updating $\eta$ is same as openai-baselines' A2C: 0.0007, training steps is 256 and the network architecture can be found in our code file "policy.py".

Figure(3) is the MI visualization in game Qbert. Note that there is a certain degree of fluctuations in the curve. This is because that unlike supervised learning, the distribution of datasets and learning signals $R^\pi(X)$ keep changing in reinforcement learning: $R^\pi(X)$ changes with policy $\pi$ and when $\pi$ gets better, the agent might find new states, in this case, $I(X, Z)$ might increase again because the agent needs to encode information from new states in order to learn a better policy. Yet finally, the MI always tends to decrease. Thus we can say that the agent in RL usually tends to follow the information E-C process.

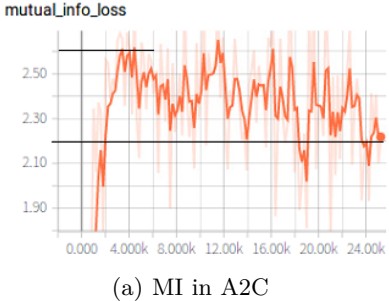 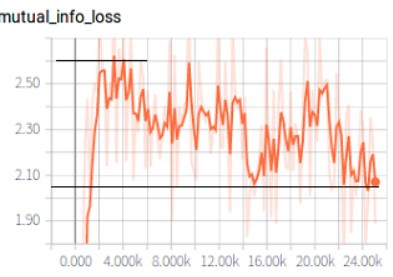

(a) MI in A2C                               (b) MI in A2C with our framework

Figure 3: Mutual information visualization in Qbert. As policy $\pi$ gets better, the agent might find new states, in this case, $I(X, Z)$ might increase again because the agent needs to encode information from new states in order to learn a better policy. Yet finally, the MI always tends to decrease. Thus it follows the information E-C process.

We argue that it's unnecessary to compute $I(Z, Y)$ like (Shwartz-Ziv & Tishby, 2017): According to (3), if the training loss continually decreases in supervised learning(Reward continually increases as showed in figure(2)(a) in reinforcement learning), $I(Z, Y)$ must increase gradually.

We also add some additional experimental results of MI visualization in the appendix(A.7).

A.7  Additional experimental results of performance and MI visualization

This section we add some additional experimental results about our framework.

Notice that in game MsPacman, performance of A2C with our framework is worse than regular A2C. According to the MI visualization of MsPacman in figure(5)(b), we suspect that this is because A2C with our framework drops the information from inputs so excessively that it misses some information relative to the learning process. To see it accurately, in figure(5)(b), the orange curve, which denotes A2C with our framework, from step(x-axis) 80 to 100, suddenly drops plenty of information. Meanwhile, in figure(4)(b), from step(x-axis) 80 to 100, the rewards of orange curve start to decrease.

As showed in figure(6), unlike Pong, Breakout, Qbert and some other shooting games, the frame of MsPacman contains much more information related to the reward: The walls, the ghosts and the tiny beans everywhere. Thus if the agent drops information too fast, it may hurt the performance.

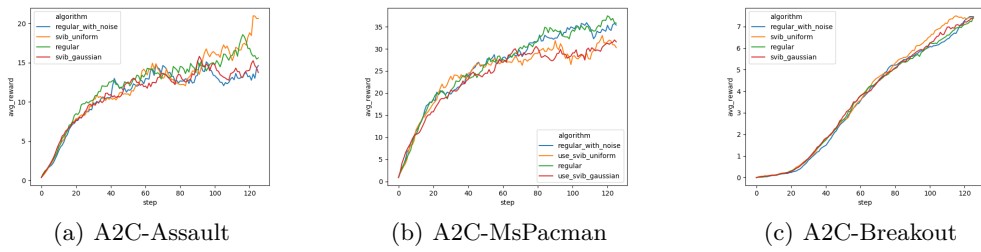

(a) A2C-Assault      (b) A2C-MsPacman      (c) A2C-Breakout

Figure 4: Additional results of performance: Average rewards over 10 episodes

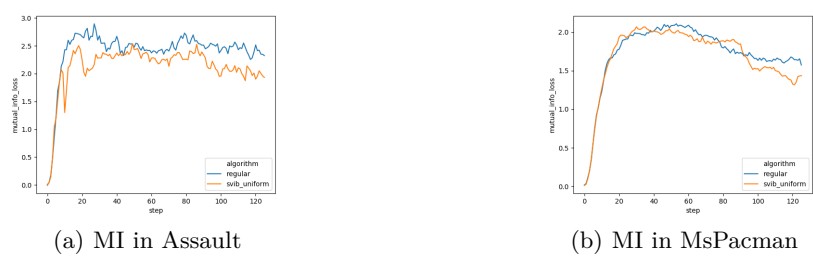

(a) MI in Assault      (b) MI in MsPacman

Figure 5: Additional results of MI visualization

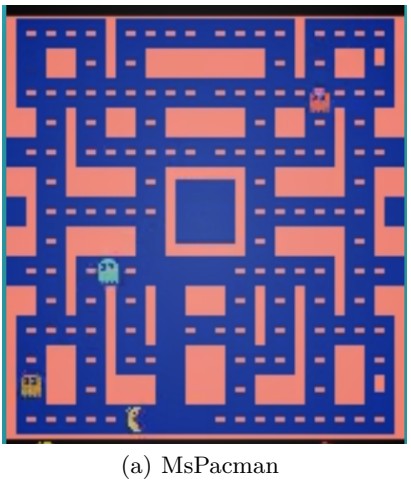

(a) MsPacman

Figure 6: The raw frame of MsPacman

