# OpenReview forum: "Learning Representations in Reinforcement Learning: an Information Bottleneck Approach"
_ICLR.cc/2020/Conference — Reject_

### Official Review · AnonReviewer3 · 2019-10-22
**Official Blind Review #3**

**Rating:** 6

**Review:**

This paper investigates the information bottleneck approach in learning representations in reinforcement learning. The authors propose 1) combining actor-critic objective and IB loss; 2) deriving optimal target distribution of the composite objective; 3) optimizing a lower bound of target function and using SVGD for optimization; 4) empirically show that the proposed method achieves promising results; 5) empirically show that there exists E-C process in RL; 6) discuss the relationship with MINE.

Pros:
1. Well organized paper with a clean presentation.
2. The technical contribution is enough (composite loss, optimal target distribution, using SVGD to solve lower bound, comparing with MINE)
3. The experimental results are promising.

Cons/Questions:
1. In Sec A.2, deriving the target distribution by KKT condition seems more rigorous (P_phi should be non-negative and sum up to 1). Using gradient equal to zero gets the same results is because the constraints are not active here (the optimal solution is not on the boundary of simplex).

2. If not use the lower bound of the target distribution (use P rather than U), is there any method to solve this?

Overall, I found this paper is interesting, with good theoretical contributions and promising empirical results.

**Experience Assessment:**

I have published one or two papers in this area.

**Review Assessment: Checking Correctness Of Derivations And Theory:**

I carefully checked the derivations and theory.

**Review Assessment: Checking Correctness Of Experiments:**

I carefully checked the experiments.

**Review Assessment: Thoroughness In Paper Reading:**

I read the paper thoroughly.

---

> ### Author Response · Authors · 2019-11-15
> **Some responses**
>
> Thanks for your review. Here are the respones of the two problems.
>
> 1.  In Sec A.2, deriving the target distribution by KKT condition seems more rigorous (P_phi should be non-negative and sum up to 1). Using gradient equal to zero gets the same results is because the constraints are not active here (the optimal solution is not on the boundary of simplex).
>
> This is a good question. We should notice that using functional derivation is just a rough way to get a usable analytical solution. Until here there is no guarantee that the result is correct. So here comes the theory 2, in this theory, according to the proof in appendix A.3, P_phi is non-negative and sum up to 1. And we proof that this P_phi is just the result we want.
>
> 2.If not use the lower bound of the target distribution (use P rather than U), is there any method to solve this?
>
> We derive an algorithm that does not need a lower bound in appendix A.5 based on our theory. Yet practical implementations and experiments will be left as a future work.

---

### Official Review · AnonReviewer1 · 2019-10-22
**Official Blind Review #1**

**Rating:** 3

**Review:**

In this paper, the authors proposed to utilize variational lower bound of mutual information for learning representations in Reinforcement learning. To optimize the proposed variational lower bound with a more flexible encoding network, the author proposed to utilize stein variational gradient descent (or amortized svgd). Instead of learning the representation separately, the authors incorporate the framework into PPO or A2C, which yields a joint training framework for policy optimization.

The authors discussed the information bottleneck in the setting of RL thoroughly, and proposed a novel target Y (\log P) in the scenario of policy optimization. Further the authors derive the variational lower bound under the setting of RL, and proposed to utilize amortized SVGD to learn the proposed variational lower bound. The authors also examine the mutual information using recent proposed MINE, which also indicates the effectiveness of the proposed method.

Utilizing recent advanced techniques in representation learning and applying it to RL tasks which have high dimensional images state spaces can accelerate the training processing. Though the problem that the authors studied is important, the proposed method lacks novelty and has not been well evaluated comparing with other representation techniques. Specifically, I have several concerns about the paper.

- Variational lower bound for RL. I admit the proposed \log p(y|z) is novel (although heuristic), but the rest of the results can be easily derived from variational lower bound framework. I suggest the author clarify and discuss about the derivation of this part with techniques introduced in variational bounds of mutual information (Poole et al. 2019).

- The derivation of updating of $\phi$ (Equation 18). The derivation is totally unnecessary since it is exactly the same as the updating formulation of Amortized SVGD (Feng et. al 2017). See Sec 3 in Feng et. al 2017 for details. I think the author should motivate the usage of implicit models (the embedding function $\phi(x, \epsilon)$) and then show that we should optimize the implicit encoders using amortized SVGD.

- The motivation of using amortized SVGD is unclear. If the authors parameterized the embedding function $\phi(z)$ with a tractable density model such as Gaussian distributions, it is still possible to optimize the proposed lower bound. Ablation studies should be conducted to demonstrate why we can enjoy the benefits of implicit models in the setting of RL.

- Baselines including other representation learning methods should be compared. The authors only compared with naive A2C or PPO, while there are many other methods which can be directly utilized to learn the representation during the policy optimization process. One example is  Contrastive Predicting Coding (Oord et. al 2018) (I noticed that the author mentioned MINE does not work well in this setting, but at least the variational lower bound using explicit models such as Gaussian encoders should be compared).

- (Minor things) The plots in the figures are hard to read.  Figure (2) shows the advantages of the proposed methods which are really hard to read and distinguish several proposed methods. It would be nice to have a table to summarize the final performance (From the plots It is hard to conclude which one is the best). In addition, figure 1&3 is unreadable (Taking screenshots of tensorboard plots is really bad).

Overall I think the paper studied an interesting problem while the motivation and the advantage of the proposed method are still unclear, which requires more discussion and comparison with other methods which can be utilized for representation learning.


Reference Papers:

Poole, Ben, et al. "On variational bounds of mutual information." arXiv preprint arXiv:1905.06922 (2019).

Feng, Yihao, Dilin Wang, and Qiang Liu. "Learning to draw samples with amortized stein variational gradient descent.” UAI 2017.

Oord, Aaron van den, Yazhe Li, and Oriol Vinyals. "Representation learning with contrastive predictive coding." arXiv preprint arXiv:1807.03748 (2018).



**Experience Assessment:**

I have published one or two papers in this area.

**Review Assessment: Checking Correctness Of Derivations And Theory:**

I assessed the sensibility of the derivations and theory.

**Review Assessment: Checking Correctness Of Experiments:**

I assessed the sensibility of the experiments.

**Review Assessment: Thoroughness In Paper Reading:**

I read the paper at least twice and used my best judgement in assessing the paper.

---

> ### Author Response · Authors · 2019-11-15
> **Some respones**
>
> Thanks for your detailed reviews! Due to recent bussiness, we are not able to respond all the problems. Yet we will augment our paper on experiments based on your advice. Here are responses to some main concerns.
>
> 1.About variational lower bound and stein variational gradients
> Actually our main contribution is constructing the theories and formulations of information bottleneck in reinforcement learning(See apendix A.1-A.3 and A.5, or typically, near-optimality theory, derivation of target distribution, improvement theory of this target distribution). Based on these theories, we propose two algorithms to optimize this problem.
> The first method is to utilize the variational lower bound and stein variational gradients, these two powerful tools themselves are not our contribution, we just combine them and create an optional new method to optimize our problem. Therefore, we just write what we think are necessary derivations about variational lower bound and amortized SVGD to introduce our optimization methods. Later on we will add citations of  (Poole et al. 2019) and Amortized SVGD (Feng et. al 2017). Sorry that we haven't heard Amortized SVGD before, all of the knowledge we know about SVGD is from three papers that we have already cited in the third part of Framework.
>
> 2.About assuming representation variable Z as Gaussian.
> The reason why we didn't take this into consideration at first is because that in the original paper of MINE, using MINE to optimize the information bottleneck in supervised learning has already outperformed the variational bottleneck(assuming Z as a Gaussian). Yet in reinforcement learning(Atari games), MINE did not even converge in our experiments. You can see appendix A.5 for detailed descriptions. So we suspect that directly using variational bottleneck might even hurt the performance. But you are definitely right, we should add the experiments to verify the performance of variational bottleneck in reinforcement learning.
>
> Actually, to the best of our knowledge, there  are no previous works that derictly apply information bottleneck in reinforcement learining(see the related work part). Thus we suspect that many general information bottleneck techniques in supervised might not work in RL. That's why we want to rigorously study the information bottleneck problem in RL.

---

> > ### Comment · AnonReviewer1 · 2019-11-15
> > **Clarifications Based on your Response**
> >
> > Thanks the author for the thorough response. Here are several comments based on your response.
> >
> > - 1. Clarification on SVGD.
> > The two papers you mentioned (Liu & Wang 2017 and Liu et. al 2017) are based on SVGD, which requires multiple particles to approximate the posterior distributions, which cannot be directly used to train a neural network. In Haarnoja et.al 2017, the paper mentioned amortized SVGD (Wang & Liu, 2017) which is an unpublished version of Amortized SVGD.
> >
> > Personally I think it will make the svgd part more rigorous if the related works are cited properly.
> >
> >
> > -2. Other representation learning methods.
> >
> > I agree that studying the information bottleneck in reinforcement learning is quite interesting, which makes the work valuable to the community. However, I still think it is necessary to compare related method for representation learning, which will make the paper more convincing.
> >
> > Overall I think the paper may still require additional work to make it publishable in the future.

---

### Official Review · AnonReviewer2 · 2019-10-23
**Official Blind Review #2**

**Rating:** 3

**Review:**

This paper proposes a representation learning algorithm for RL based on the Information Bottleneck (IB) principle. This formulation leads to the observed state X being mapped to a latent variable Z ~ P(Z | X), in such a way that the standard loss function in actor-critic RL methods is augmented with a term minimizing the mutual information between X and Z (which can be seen as a form of regularization). This results in a loss that is difficult to optimize directly in the general case: the authors thus propose to approximate it through a variational bound, using Stein variational gradient descent (SVGD) for optimization, which is based on sampling multiple Z_i’s for a given state X, so as to compute an approximate gradient for the parameters of the function mapping X to Z. Experiments show that when augmenting the A2C algorithm with this technique, (1) the mutual information I(X, Z) decreases more quickly (better « compression » of the information), and (2) better sample efficiency is observed on 5 Atari games (with also encouraging results with PPO on 3 Atari games).

In spite of the interesting theoretical contributions, I have to recommend rejection as the current empirical evaluation of the proposed approach is extremely limited, making it difficult to assess its benefits over more straightforward algorithms.

On the positive side, the authors derive a sensible approach to IB representation learning in RL, and provide solutions to the optimization challenges it leads to. I did not have time to check all the maths in the Appendix (I only went through the derivations in A.1 and A.2), but they seem to make sense overall (though it is unclear to me if the new algorithm proposed in A.5 is a practical one, so I am not taking it into account in this evaluation).

The key negative point is definitely the weak empirical evaluation. The main results are from a limited sample of 5 Atari games, when the full Atari benchmark has 10x more games and is known to exhibit high variance among games when comparing RL algorithms (the additional results from the Appendix on 3 additional games also show situations where the proposed method does not seem to help much, confirming that larger scale experiments are needed for a proper evaluation). In addition it seems like each algorithm is run only once (instead of using multiple seeds) and only over ~200K timesteps, which is three orders of magnitude lower than results typically reported on Atari. Another issue is that there is no comparison to other representation learning techniques (like those mentioned in the related work section, or the recent "Unsupervised State Representation Learning in Atari"), nor to a natural and more straightforward variant of the proposed method where Z would simply be sampled from a (learned) Gaussian distribution Z ~ N(mu(X), var(X)), which at first sight seems like an easier-to-optimize objective (using the reparameterization trick)… I may be wrong, but then this should probably be explained in the paper (I realize that the proposed approach is more general, but then it should be shown how this extra flexibility can lead to improved results). Finally, the impact on runtime performance is not analyzed: how much slower do A2C / PPO become when optimizing the mutual information term with SVGD? Overall it is really unclear that better RL results can be obtained through this technique.

Another important issue is that I found the paper rather difficult to follow, due to some inconsistent / unclear notations or equations. Here are the main ones I noted:
•	The discount factor is not accounted for in the derivation of the objectives in eq. 1-2 (I know this is often the case in practice, but the reason for dropping it should at least be mentioned)
•	The jump from V^pi(X) to V^pi(Z) at the end of Section 3 is explained too succintly. It suggests that V^pi(Z) must be constant (equal to V^pi(X)) over all values of Z that may be sampled from X, which as far as I can tell is not the case in the rest of the paper. It is also unclear whether pi depends on Z or X. And the notation J(Z) makes it look like J does not depend on X, but it seems like it does because even if pi depends only on Z, by its definition V^pi(Z) actually still depends on X (this also leads to weird equations like eq. 33 where X does not appear in the right-hand side). Overall this is rather confusing.
•	The « for every X » at the top of p. 4 does not make sense to me, due to the term I(X, Z) in eq. 3 where X is a random variable and thus does not take a specific value.
•	The paragraph below eq. 4 is a bit confusing. It looks like it amounts to saying that Y is a Gaussian around V(Z)?
•	Eq. 5 suggests that R depends on Z, but shouldn’t it depend on X? If it depended on Z, then wouldn’t it influence the optimization since by modifying P(Z|X) we can control the distribution on Z and thus the distribution on R?
•	In eq. 10 it is unclear whether L1 and L2 contain the expectation, also L2 is defined as a function of both theta and phi but seems to depend only on theta
•	Below eq. 15 it is said that « P is the distribution of Z » but P does not appear in eq. 15
•	In eq. 16 should the phi on the left hand side be phi* as in eq. 15?
•	Below eq. 16 it is said « Notice that C has been omitted », but it is unclear whether it was not included to alleviate notations or because it disappears naturally in the mathematical derivation of eq. 16
•	The motivation for introducing zeta below eq. 17 is unclear, especially since it seems to play an important role considering that zeta = 0.005 << 1 is used in the experiments (with no explanation as to how this specific value was selected)
•	\hat{L}(Z_i, theta, phi) (between eq. 17 and 18) does not seem to be defined
•	What is the motivation for using the Gaussian U(Z) as described in Section 5? In particular I find it weird that it depends on X_i, while U(Z) is supposed to replace the marginal P(Z) and not the conditional P(Z | X)

Minor points:
•	In the definition of Y_t = R_t = sum_i=0^n-2 … above eq. 4, I think the sum should be up to n-1 for an n-step return
•	Eq. 4 uses R_t on the right hand term instead of Y_t which looks weird
•	Theorem 1 states « Assume that for any epsilon > 0, … »: « for any » should probably be « there exists », since if the inequality was true for any epsilon, it would imply both mutual informations are equal (also the formulation of the theorem does not make it clear that the last inequality is the main result)
•	Footnote 1 p. 4: I(X, Y) should be I(X, Z)
•	In eq. 15 the phi below the argmax should be in bold
•	Please add a reference that the reader can refer to in order to understand where eq. 20 is coming from
•	« Apparently » is used in a couple of places but should probably be replaced with another word
•	When U is uniform, how do you choose its support?
•	After « median of pairwise distances between the particles », I think i=1 should be j=1
•	It is unclear to me what « A2C with noise is »: it is said that the same phi(X, epsilon) is used « as A2C with our framework », but is it the same phi as A2C with uniform SVIB or A2C with Gaussian SVIB? And whichever it is, how does it differ from the one it is equal to?
•	« we add 21 to all four curves in order to make exponential moving average »: I do not understand that sentence
•	« we set the number of samples as 26 for the sake of computation efficiency »: I fail to see how going from 32 to 26 is going to make a major difference in computational efficiency

Update: thank you for your response, but in the absence of a revised version addressing my concerns (as well as those from the other reviewers), I cannot increase my rating

**Experience Assessment:**

I have read many papers in this area.

**Review Assessment: Checking Correctness Of Derivations And Theory:**

I assessed the sensibility of the derivations and theory.

**Review Assessment: Checking Correctness Of Experiments:**

I carefully checked the experiments.

**Review Assessment: Thoroughness In Paper Reading:**

I read the paper thoroughly.

---

> ### Author Response · Authors · 2019-11-15
> **Some responses to some main concerns**
>
> Thanks for your detailed reviews! Due to recent bussiness, we are not able to respond all the concerns. Yet we will augment our paper both on experiments and notations based on your detailed and careful advice. Again many thanks to your careful reading. Here are responses to some main concerns.
>
> 1.About seeds
> Due to the limitations of compution resources and time, we are only able to run 1 seed for each game before submission. Yet these days we are working on adding multi-seeds experiments(3 seeds specifically), now we have the results that our algorithm performs better than original A2C in five games(Pong, Assault, BeamRider, Qbert, AirRaid), more games are running now.
> 2.About timesteps
> We are not sure about if we are wrong: Since our batchsize is 64, our total frames ought to be 200k * 64 * 4 = 51,200,000 frames. This is the same setting as openai-baselines. Emmm, do you think these timesteps are still not enough for evaluation?
> 3.About assuming representation variable Z as Gaussian.
> The reason why we didn't take this into consideration at first is because that in the original paper of MINE, using MINE to optimize the information bottleneck in supervised learning has already outperformed the variational bottleneck(assuming Z as a Gaussian). Yet in reinforcement learning(Atari games), MINE did not even converge in our experiments. You can see appendix A.5 for detailed descriptions. So we suspect that directly using variational bottleneck might even hurt the performance. But you are definitely right, we should add the experiments to verify the performance of variational bottleneck in reinforcement learning.
>
> We are really grateful that you can give us the directions to augment our experiments. Your advice of notation is also valuable. After we finish the current bussiness, we will make a more rigorous correction on the notations.

---

### Decision · Program_Chairs · 2019-12-19

**Decision:**

Reject

**Comment:**

The authors propose to use the information bottleneck to learn state representations for RL. They optimize the IB objective using Stein variational gradient descent and combine it with A2C and PPO. On a handful of Atari games, they show improved performance.

The reviewers primary concerns were:
*Limited evaluation. The method was only evaluated on a handful of the Atari games and no comparison to other representation learning methods was done.
*Using a simple Gaussian embedding function would eliminate the need for amortized SVGD. The authors should compare to that alternative to demonstrate the necessity of their approach.

The ideas explored in the paper are interesting, but given the issues with evaluation, the paper is not ready for acceptance at this time.